# Predicting the Potential Distribution of Non-Native Mammalian Species Sold in the South African Pet Trade

Ndivhuwo Shivambu, Tinyiko C. Shivambu and Colleen T. Downs *

DSI–NRF Centre for Excellence in Invasion Biology, and Centre for Functional Biodiversity, School of Life Sciences, University of KwaZulu–Natal, Private Bag X01, Scottsville, Pietermaritzburg 3209, South Africa; ndivhuwomaligana@gmail.com (N.S.); shivambucavin@gmail.com (T.C.S.)
* Correspondence: downs@ukzn.ac.za

**Abstract:** The pet trade is one of the main pathways of introduction of several mammals worldwide. In South Africa, non-native mammalian species are traded as pets, and so far, only four of these species are considered invasive. We used a list of 24 companion mammalian species compiled from a previous study. We selected a subset of 14 species for species distribution modeling (SDM) based on their trade popularity, invasion history and potential economic and socio-economic impacts. We aimed to estimate their potential distribution using their distribution records. Our SDM indicated that climate in South Africa was suitable for most traded species. However, commonly and easily available species had the broadest areas of suitable climates, such as house mice (*Mus musculus*) and Norwegian rats (*Rattus norvegicus*). In addition, the model with a human footprint suggested a high risk of invasion for Norwegian rats but less for house mice distribution. This assessment suggests the need of strict trade regulations and management strategies for pet mammals with broader suitability, which are already invasive, and most available for sale. In addition, our results provide a baseline approach that can be used to identify mammalian pet species with a potential risk of invasion so that urgent preventive measures can be implemented.

**Keywords:** human footprint; species distribution modeling; invasive species; introduction pathway; impact

## 1. Introduction

Several mammalian species have been introduced in South Africa and other countries for different purposes, including pest control, research, food, fur markets, game, hunting, zoo, and as pets [1–6]. Mammalian species are among the most successful invaders worldwide, and their success as biological invaders has mostly been linked to their ability to breed successfully, extensive physiological tolerance, association with humans, broad habitats, and diets [1,7–9]. Invasive mammalian species are associated with negative impacts on agriculture, human health, infrastructure, native fauna and biota in general [9–13].

It is vital to investigate the invasion history and potential distribution of non-native species to prevent them from becoming invasive and causing impacts. Studies have suggested that matching the climate between the native and non-native areas of a species is essential in identifying the invasion potential for a species [14,15]. Species distribution modeling (SDM) is a widely used tool to predict potentially suitable areas where non-native species may establish and become invasive if introduced into favorable environments [16–18].

Species distribution modeling is also known as a bioclimatic envelope, ecological niche modeling or habitat suitability modeling, which uses an organism's occurrence records combined with geographical and environmental variables to predict species suitability [19–23]. The SDM has been applied in a range of fields, including biodiversity conservation and wildlife management [22,24,25], climate change [26], species extinction assessment [27], risk assessment [28–30] and effects of human footprint [31,32]. Distribution modeling can

also be used to develop and implement early detection, warnings and prevent potential invaders [31,32]. Some studies have suggested that invasive species in some parts of the world are likely to become invasive in other regions, given that these areas have similar environmental suitability [14,33]. Additionally, socio-economic factors such as human population density, cropland, built environments, pasture land, railways, night-time lights, roads, and maneuverable waterways have been responsible for the invasion of several species [34–36]. For example, the invasion success of the commensal rodents such as house mice (*Mus musculus*) and Norwegian rats (*Rattus norvegicus*) [37–41].

Several non-native mammalian species introduced through the pet trade have established feral populations outside their native ranges, e.g., sugar gliders (*Petaurus breviceps*) and common marmosets (*Callithrix jacchus*) [10,42]. The sale of non-native species typically remains unregulated in many countries, leading to many species being translocated between the regions, sometimes resulting in pet releases or escapes [43–46]. Several non-native mammals, including the world's worst invasive species, such as house mice and Norwegian rats, are sold in the South African pet trade [47]. Previous studies focused on determining the extent of the mammalian species trade and assessing their potential environmental and socio-economic impacts in South Africa [13,48]. Relatively little has been done to investigate whether species sold in South Africa have potential climatic suitability. Therefore, we compiled a list of non-native mammalian species sold as pets in South Africa obtained from a previous study [48] to determine their potential distribution based on ecological niches in South Africa. We also determined if the human footprint explained the potential distribution of house mice and Norwegian rats. We predicted that species with high availability, a history of invasion elsewhere, and worldwide occurrence records would have greater invasion potential. In addition, we expected the human footprint to influence the potential distribution of house mice and Norwegian rats.

## 2. Materials and Methods

### 2.1. Data Collection and Species Selection

For this study, we used a list of traded non-native mammalian pet species compiled from a previous study [48]. Species recorded were compiled by visiting a total of 122 pet stores (Figure 1) from nine South African provinces between September 2018 and September 2019 [48]. The study included small- and medium-sized mammalian species < 5 kg, belonging to the following orders: Rodentia, Carnivora, Primates, Eulipotyphla, Lagomorpha and Diprotodontia [48]. We selected species from the previous study [48] for SDM based on the following: (1) potential environmental and socio-economic impacts [13], (2) distribution records in both or either native or introduced ranges, (3) history of invasion elsewhere and introduction pathway, and (4) availability (popularity) in the South African pet trade [12,48] (Table 1). To evaluate species availability in the pet trade, we followed criteria from Chucholl [49]: (i) "very rare", species available for a short period in either one source of trade (online or pet store), less than four provinces or less than three online platforms and in low quantity; (ii) "rare", species occasionally available in either one source of trade, few provinces or online platforms and in low quantity; (iii) "common", species frequently available in either one source of trade, more provinces or online platforms and high quantity, and (iv) "very common", species always available in all the sources of trade, more than four provinces or more than three online platforms and high quantity.

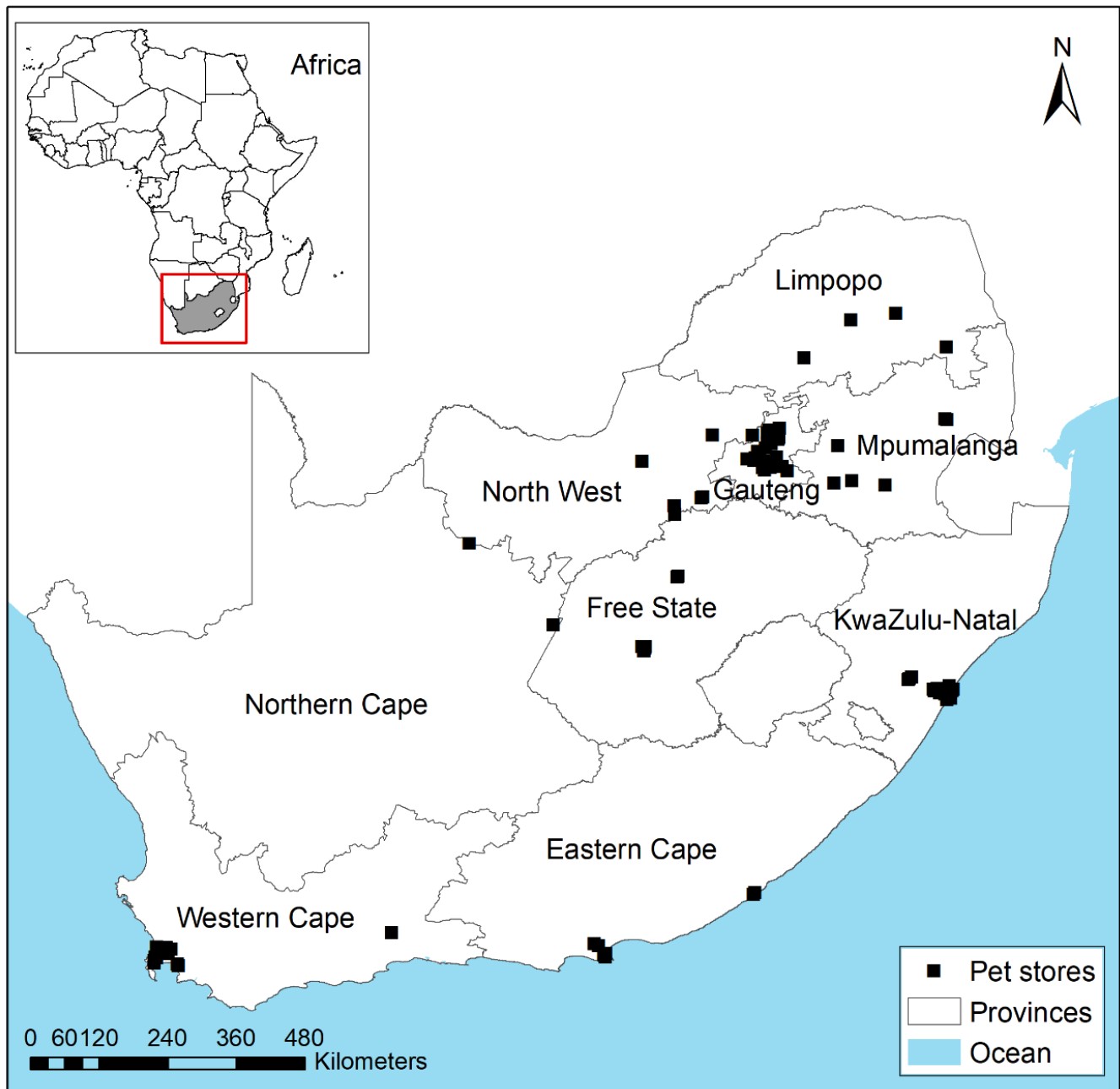

**Figure 1.** A map showing localities of pet stores where non-native mammalian species were recorded sold in South Africa between September 2018 and September 2019 (data from Shivambu et al. [48]).

**Table 1.** List of non-native pet mammalian species selected for SDM with their data on their availability in the pet trade, native ranges, status in South Africa, invasion history and introduction pathways based on previous studies [13,48]. Abbreviations for areas or countries are given in the footnote.

| Order | Scientific Name | Common Name | Species Availability | | Native Area | Status in South Africa | Countries Introduced | Introduction Pathways |
|---|---|---|---|---|---|---|---|---|
| | | | No. of Pet Store | No. of Online Websites | | | | |
| Rodentia | *Cavia porcellus* | Guinea Pig | 70 | 3 | SA | Captivity | Not invasive | Not invasive |
| | *Meriones unguiculatus* | Mongolian gerbil | 10 | 2 | MO, NECN | Captivity | Not invasive | Not invasive |
| | *Mus musculus* | House mouse | 68 | 2 | E | Invasive | All continents except AN | Accidental escape (hitchhikers on trading ships and cargos) [1] |
| | *Mesocricetus auratus* | Golden hamster | 54 | 3 | SY, TR, GR, RO, BE | Captivity | Not invasive | Not invasive |
| | *Phodopus sungorus* | Winter white dwarf hamster | 59 | 3 | MO, NECN | Captivity | Not invasive | Not invasive |
| | *Rattus norvegicus* | Norwegian rat | 78 | 4 | CN, RU, JP | Invasive | All continents except AN | Accidental escape (hitchhikers on trading ships and cargos) [1] |
| | *Sciurus carolinensis* | Eastern gray squirrel | 0 | 1 | ENA | Invasive | RSA, IE, IT, UK, NA | Intentional release and accidental escape (pet, ornamentation) [1,50] |
| Carnivora | *Mustela putorius furo* | Domesticated ferret | 2 | 2 | WE, NMR | Captivity | RAA, UK, NZ | Intentional release and accidental escape (pet, hunting, fur farming) [1,51,52] |
| Diprotodontia | *Petaurus breviceps* | Sugar glider | 3 | 3 | AU, PNG | Captivity | Tas | Accidental escape (pet) [53,54] |
| Eulipotyphla | *Atelerix albiventris* | African pygmy hedgehog | 29 | 6 | EAF | Captivity | Not invasive | Not invasive |
| Lagomorpha | *Oryctolagus cuniculus* | European rabbit | 101 | 3 | EU | Invasive | All continents except AN | Intentional release and accidental escape (food or farming) [1,55] |
| Primates | *Callithrix jacchus* | Common marmoset | 2 | 4 | ECBR | Captivity | SEBR, NEBR | Release and escape (pet) [42,56] |
| | *Callithrix penicillata* | Black-tufted ear marmoset | 3 | 2 | ECBR | Captivity | SEBR | Release and escape (pet) [42,57] |
| | *Saimiri sciureus* | Common squirrel monkey | 0 | 1 | SA | Captivity | RJ | Release (pet) [42,58] |

Abbreviations for countries or areas: South America (SA), North America (NA), Mongolia (MO), North-eastern China (NECN), Eurasia (E), Syria (SY), Turkey (TR), Greece (GR), Romania (RO), Belgium (BE), West-central Chile (WCCL), China (CN), Russia (RU), Japan (JP), Eastern North America (ENA), Western Eurasia (WE), North Morocco (NMR), Australia (AU), Papua New Guinea (PNG), Eastern Africa (EAF), Europe (EU), East-central Brazil (ECBR), Antarctica (AN), South Africa (RSA), Ireland (IE), Italy (IT), United Kingdom (UK), Azores (RAA), New Zealand (NZ), Tasmania (TAS), Southeast Brazil (SEBR), Northeast Brazil (NEBR), Rio de Janeiro (RJ).

## 2.2. Species Occurrence Data, Model Fitting, Prediction and Evaluation

We downloaded current global spatial occurrence records (i.e., invaded and native) for the 14 mammalian species from the Global Biodiversity Information Facility (GBIF) [59–72] to develop species distribution models. The GBIF comprises the largest occurrence dataset collated from observed data from different accredited sources across the world. For our study, we used GBIF museum datasets as species identification and locations were confirmed. The distribution records were assessed for quality and cleaned using the Biogeo package in R [73]. Records falling into the ocean and duplicates within a 10 min grid cell were removed, therefore leaving only one occurrence point per 10 arcmin pixel. This procedure reduced spatial bias of occurrence data which substantially improved prediction reliability [74]. A subset of between 64 and 10,672 occurrence records (Table 2) was used for modeling.

**Table 2.** Percent contribution of environmental variables for *Mus musculus* and *Rattus norvegicus* when the human footprint was added to the model. Variables contributing the most are shown in bold for each species.

| Variables | *Mus musculus* | *Rattus norvegicus* |
|---|---|---|
| Bio 2: Mean Diurnal Range (Mean of monthly (max temp–min temp)) | 2.4 | 0.9 |
| Bio 3: Isothermality (BIO2/BIO7) (×100) | **42.7** | **22.2** |
| Bio 4: Temperature Seasonality (standard deviation ×100) | 0 | 4.4 |
| Bio 5: Max Temperature of Warmest Month | – | – |
| Bio 6: Min Temperature of Coldest Month | – | – |
| Bio 7: Temperature Annual Range (BIO5–BIO6) | – | – |
| Bio 8: Mean Temperature of Wettest Quarter | 1.7 | 1.2 |
| Bio 9: Mean Temperature of Driest Quarter | **19.8** | 0 |
| Bio 10: Mean Temperature of Warmest Quarter | 0 | 8.3 |
| Bio 13: Precipitation of Wettest Month | 8.1 | 4.7 |
| Bio 14: Precipitation of Driest Month | 5.5 | 11.4 |
| Bio 15: Precipitation Seasonality (Coefficient of Variation) | 0.6 | 6.5 |
| Bio 18: Precipitation of Warmest Quarter | 0.3 | 0.8 |
| Bio 19: Precipitation of Coldest Quarter | **16.2** | **14.6** |
| Human Footprint | 2.7 | 25 |

We used the SDM package [75] in R version 3.6.1 [76] to develop ecological niche models of the selected non-native mammalian species traded in South African online and pet stores. A set of 19 bioclimatic variables (https://www.worldclim.org/ (17 June 2020), [77]) at 10 min spatial resolution was downloaded, and we used these as predictors to describe each mammalian species suitability. These variables were used because they are likely to have direct physiological and ecological processes that affect the species distribution [78,79]. We tested for correlations between bioclimatic variables using the variance inflation factor function (VIF) [80] and Pearson (r) correlation coefficients to detect collinearity. The collinear bioclimatic variables were excluded when building the model, and 11 variables were used for each species (see Table 3). In addition to the bioclimatic variables for house mice and Norwegian rats, we added a global map of human influence, the "human footprint" [36], as an additional predictor variable since the expansion of these species is favored by human presence (activity). The human footprint index was downloaded from https://sedac.ciesin.columbia.edu/data/set/wildareas-v3--2009-human-footprint (28 January 2021) [81]. The potential distribution for these two rodents was produced using ArcGIS version 10.4.1 [82].

We fitted the model using maximum entropy (Maxent) algorithm version 3.3.3.k [83]. To project potential species distribution models, Maxent requires presence and pseudo-absences records [19,75,84]. For this study, 10,000 pseudo-absences records were randomly drawn from a defined background at average runs of 100 bootstrap replications [84–88]. The occurrence data for each species were partitioned into a training and a testing dataset using *k*-fold partitioning. About 80% of the dataset was used as training and the remaining 20% as testing dataset. The convergence threshold was $1 \times 10^{-5}$ based on 10 replications, and parameters were set to 5000 iterations.

Model performance for species was evaluated using the independent-threshold statistic, AUC (Area Under Curve) of the receiver operating characteristic curve (ROC) [89]. The AUC values range from 0 to 1, with values below 0.7 considered poor, between 0.7 and 0.9 considered good, and greater than 0.9 considered excellent [89,90]. Ensembles of all the Maxent methods for each species were generated to create a consensus model among them. The potential distribution map for each species was plotted using R statistical software version 3.6.1 [76] for the analyses.

**Table 3.** Predictor variables, the percentage contribution, and AUC training values for the 14 non-native mammalian species sold as pets in South Africa. Bioclimatic variables contributing the most are shown in bold for each species, and the full description for each variable is provided in the footnote.

| Species names | Distribution records | AUC | Bio 2 | Bio 3 | Bio 4 | Bio 8 | Bio 9 | Bio 10 | Bio 13 | Bio 14 | Bio 15 | Bio 18 | Bio 19 |
|---|---|---|---|---|---|---|---|---|---|---|---|---|---|
| *Atelerix albiventris* | 284 | 0.984 | 3 | **25** | 16 | 1 | 0 | 5 | **20** | 12 | 0 | 10 | 8 |
| *Callithrix jacchus* | 310 | 0.973 | 3.5 | **21.5** | 35 | 4 | 0 | 5 | 7 | 12 | 2 | 0 | 10 |
| *Callithrix penicillata* | 301 | 0.994 | 4 | **30.6** | 0 | 6 | 12.4 | 0 | 11 | 5 | 0 | **26** | 5 |
| *Cavia porcellus* | 69 | 0.824 | 2 | **35.2** | 0 | 6 | 10.8 | 0 | 3 | **34.3** | 1 | 3.2 | 4.5 |
| *Meriones unguiculatus* | 180 | 0.761 | 8 | 7 | **39** | 0 | 0 | 20 | 2 | 1 | 3 | 5 | **25** |
| *Mus musculus* | 10,672 | 0.799 | 0 | 20.3 | **34.7** | 1 | 0 | 3 | 2 | 0 | 2 | 0 | **37** |
| *Mesocricetus auratus* | 64 | 0.862 | 1.5 | **43.7** | 0 | 2.5 | 16.3 | 0 | 4 | 2.3 | 0 | 1 | **28.7** |
| *Mustela putorius furo* | 478 | 0.968 | 13 | 6 | 0 | 1 | 0 | **19.5** | 4.5 | **44** | 10.7 | 0 | 1.3 |
| *Oryctolagus cuniculus* | 946 | 0.948 | 0 | 8 | 0 | 1 | **22** | 0 | 0 | 0 | 4 | **45** | 20 |
| *Phodopus sungorus* | 153 | 0.954 | 0 | **59** | 9 | 5 | 0 | **19.9** | 2 | 3 | 1.7 | 5.4 | 0 |
| *Petaurus breviceps* | 1000 | 0.979 | 4 | **52** | 0 | 0 | 2 | 0 | 1 | **28** | 2 | 8 | 3 |
| *Rattus norvegicus* | 2615 | 0.97 | 3 | **22** | 0 | 2 | 11 | 0 | 1 | 14.2 | 3.8 | 6.8 | **36.2** |
| *Saimiri sciureus* | 837 | 0.924 | 1 | **28** | 0 | 0 | 3 | 0 | 3.1 | 16 | 1.9 | 5 | **42** |
| *Sciurus carolinensis* | 2048 | 0.98 | 13.7 | **22.3** | 0 | 0 | 0 | 0 | 6 | **58** | 0 | 0 | 0 |

**Bio 2**: Mean Diurnal Range (Mean of monthly (max temp–min temp)), **Bio 3**: Isothermality (BIO2/BIO7) (×100), **Bio 4**: Temperature Seasonality (standard deviation ×100), **Bio 8**: Mean Temperature of Wettest Quarter, **Bio 9**: Mean Temperature of Driest Quarter, **Bio 10**: Mean Temperature of Warmest Quarter, **Bio 13**: Precipitation of Wettest Month, **Bio 14**: Precipitation of Driest Month, **Bio 15**: Precipitation Seasonality (Coefficient of Variation), **Bio 18**: Precipitation of Warmest Quarter, **Bio 19:** Precipitation of Coldest Quarter.

## 3. Results

The models performed well in estimating the potentially suitable areas for all the species, with AUC values ranging from 0.76 to 0.99 (Tables 2 and 3). The generated model for house mice with bioclimatic variables almost covered the whole country (Figure A1). However, for Norwegian rats, the model with bioclimatic variables predicted coastal parts of Western Cape Province as potentially suitable areas (Figure A1). Bio 3 and bio 19 contributed the most to the models for house mice and Norwegian rats (Table 3). Bio 3 still contributed the most to both rodents' potential distribution when the human footprint was included, indicating that it has the most useful information by itself. However, the human footprint contributed 2.7% to the potential distribution of house mice (Table 2).

The anthropogenic factors were found to have an influence on the potential distribution of Norwegian rats, with the human footprint contributing the most to the model (Table 2). In comparison with the model with bioclimatic variables, the potential distribution areas for Norwegian rats expanded to other coastal areas, as well as the inland areas (Figure 2). For house mice, the potential distribution areas shifted for some provinces when the human footprint was added. For example, KwaZulu-Natal Province distribution shifted from coastal areas to the inland (Figure 2). The models showed that these two rodents' occurrences and selling points were within the predicted suitable areas (Figure 2, Figure A1).

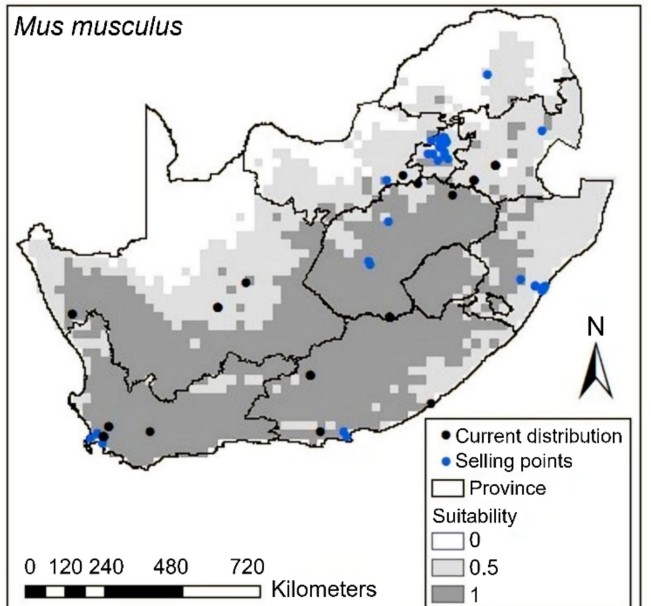
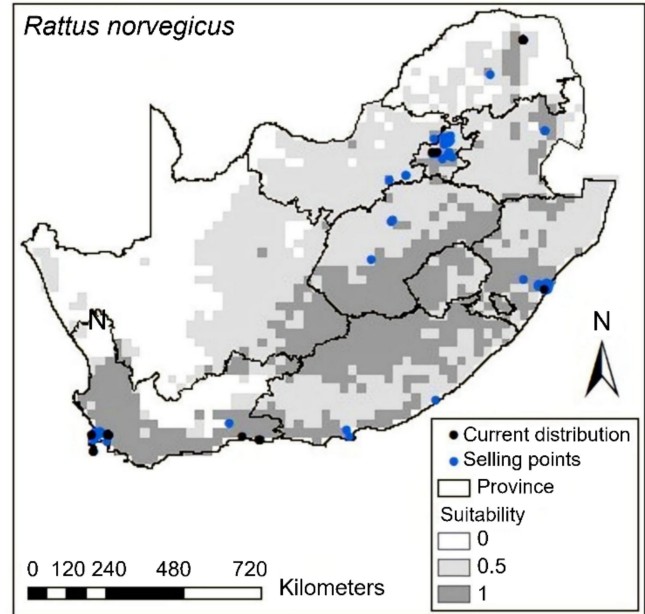

**Figure 2.** The potential distribution of house mice (*Mus musculus*) and Norwegian rats (*Rattus norvegicus*) predicted by species distribution model with the human footprint (Note: The color ramp threshold on the right measured the suitability: dark gray indicates the most suitable areas, decreasing to light gray, with white being unsuitable).

In addition to the models of the species with current South African occurrence records, European rabbits (*Oryctolagus cuniculus*) were estimated to establish in the larger parts of the Western Cape Province. Eastern gray squirrels (*Sciurus carolinensis*) were predicted to establish in small areas of the coastal regions of KwaZulu-Natal and Western Cape Provinces (Figure 3). Most of the selling points for these two species were not within the potential predicted suitable areas (Figure 3). We found that the variables that contributed the most to the model of European rabbits and eastern gray squirrels were bio 18 and bio 14, respectively (Table 3). Amongst the species with no occurrence records in South Africa, Guinea pigs (*Cavia porcellus*) had the largest potential suitable areas, with their selling points within the predicted areas (Figure 3). The areas which were predicted as suitable for this species were mainly described by bio 3 and bio 14 (Table 3).

The models generated for most species had predicted distributions in the coastal areas of South Africa, except for the Mongolian gerbils (*Meriones unguiculatus*) and winter white dwarf hamsters (*Phodopus sungorus*), which were predicted to establish Northern Cape Province inland areas (Figures 3 and 4). Temperature seasonality and precipitation of the coldest quarter explained most of the potential distribution of Mongolian gerbils. The distribution of winter white dwarf hamsters was mainly described by isothermality (Table 3). In the case of the primates, the model predicted that common marmosets and black-tufted ear marmosets (*Callithrix penicillata*) could occupy the same coastal areas, while common squirrel monkeys (*Saimiri sciureus*) could establish in the small coastal parts of the Eastern Cape Province. The bioclimatic variables contributing the most to the potential distribution of these primates were temperature seasonality, isothermality, and precipitation of the coldest quarter (Table 3).

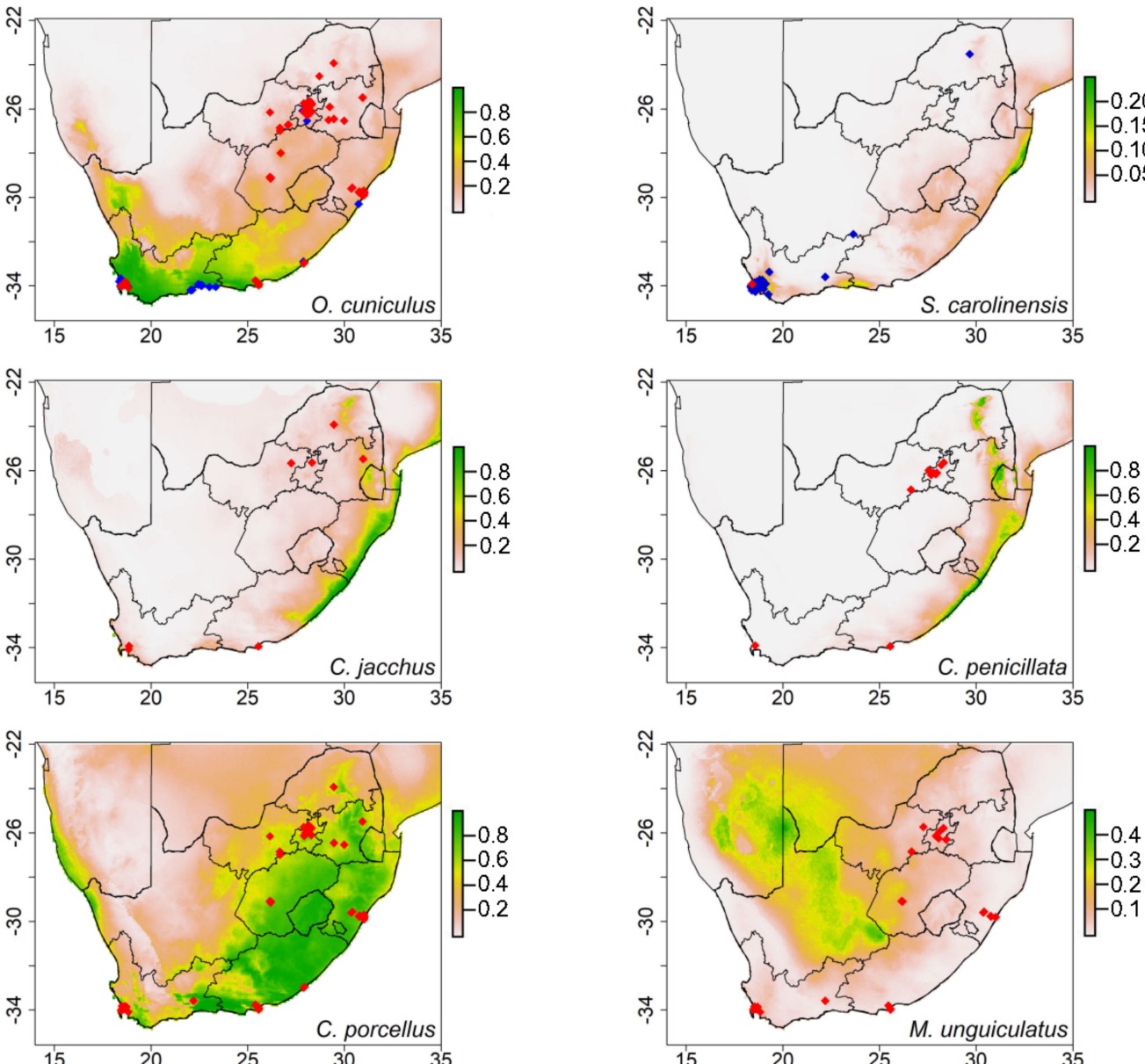

**Figure 3.** The potential distribution from ecological niche modeling showing the potential suitability for European rabbits (*Oryctolagus cuniculus*), eastern gray squirrels (*Sciurus carolinensis*), common marmosets (*Callithrix jacchus*), black-tufted ear marmosets (*Callithrix penicillata*), Guinea pigs (*Cavia porcellus*), and Mongolian gerbils (*Meriones unguiculatus*) in South Africa (Note: The color ramp threshold on the right measured the climatic suitability: green indicates the most climatic suitable areas, decreasing to yellow and orange, with light gold and white being unsuitable). Blue dots indicate current distribution records, and red dots indicate selling points.

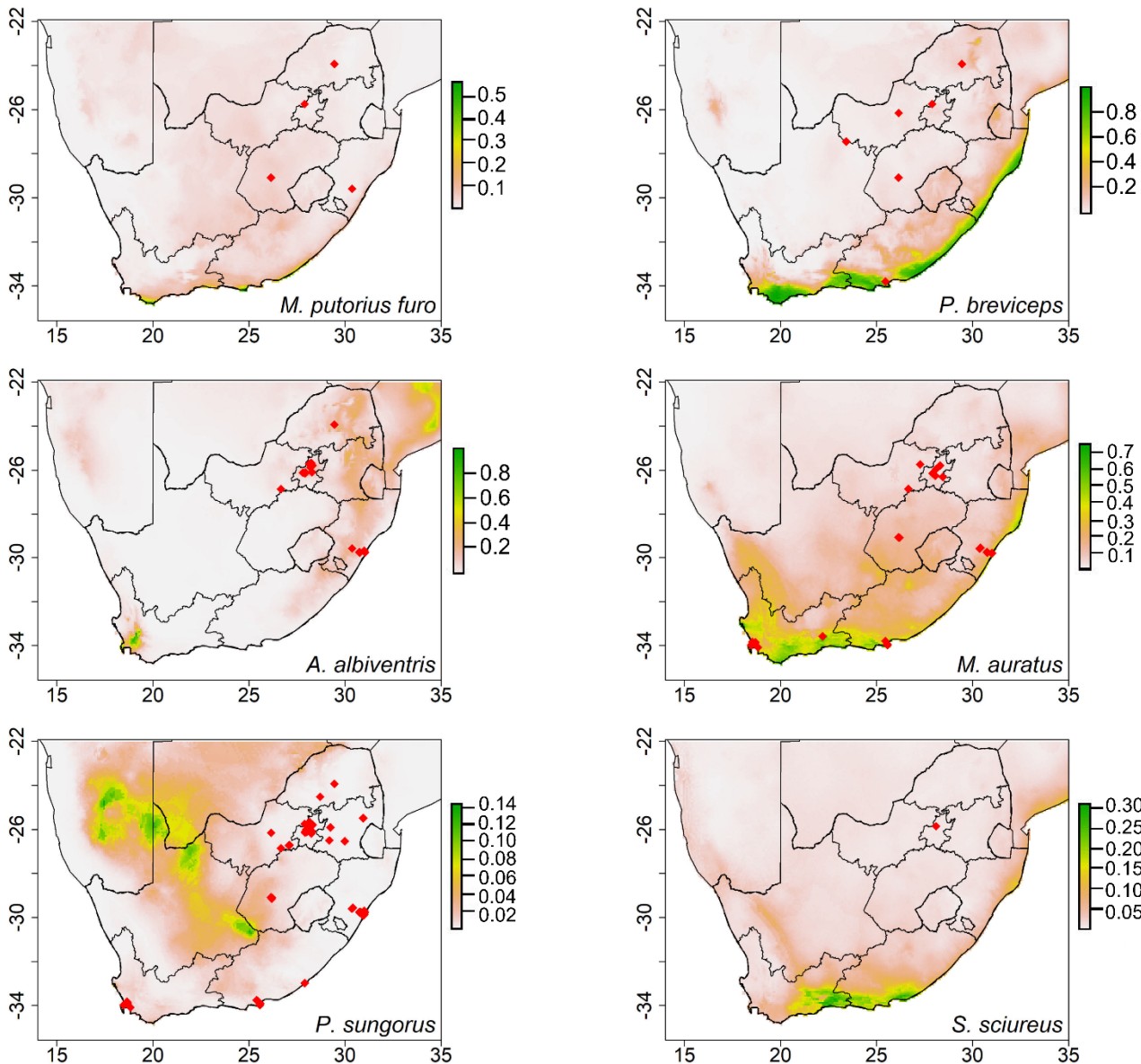

**Figure 4.** The potential distribution maps showing the areas that are potentially climatically suitable for domesticated ferrets (*Mustela putorius furo*), sugar gliders (*Petaurus breviceps*), African pygmy hedgehogs (*Atelerix albiventris*), golden hamsters (*Mesocricetus auratus*), winter dwarf hamsters (*Phodopus sungorus*) and common squirrel monkeys (*Saimiri sciureus*) in South Africa. (Note: The color ramp threshold on the right measured the climatic suitability: green indicates the most climatic suitable areas, decreasing to yellow and orange, with light gold and white being unsuitable). Red dots indicate selling points.

Species with the smallest predicted suitable areas included domesticated ferrets (*Mustela putorius furo*) and African pygmy hedgehog (*Atelerix albiventris*) (Figure 4). Bio 14 and bio 3 seemed to describe the potential distribution of these species (Table 3). The generated model for sugar gliders presented potential distribution in the coastal areas of South Africa (Figure 4). For golden hamsters (*Mesocricetus auratus*), the models estimated potential areas in both the inland and coastal areas of the Western Cape and Eastern Cape Provinces (Figure 4). This species could potentially occupy small parts of the Northern Cape Province (Figure 4). Isothermality contributed the most to the potential distribution of both sugar gliders and winter dwarf hamsters (Table 3).

## 4. Discussion

Several species with distribution records in the present study are invasive in various countries, and the introduction of some of these species resulted from accidental escape and intentional releases from urban areas (Table 1). Cities and towns from which the potential climatic suitability of the studied species matched are considered to be at risk of becoming invaded as many introductions take place in the urban areas where pet stores are situated and where there is high human population density [15,91]. In addition, urban areas are regarded as hotspots of biological invasion as many introductions start there [92]. Additionally, some species' selling points were within the projected highly suitable areas; consequently, such species may become invasive if released or if they escaped captivity.

As predicted, our study found that very common mammalian species in the South African pet trade had larger climatic suitability than rare species. These included Guinea pigs, winter white dwarf hamsters, golden hamsters, house mice, and Norwegian rats. The latter two species have been introduced on all the continents except Antarctica and are already invasive in South Africa and its offshore islands [6,93,94]. In urban areas, they are found in populated built areas such as townships, especially in the cities such as Durban, Johannesburg and Pretoria ([95–97], pers obs.). The two species are considered highly invasive and are difficult to control due to their high reproductive potential and broad diet [98]. The breeding of these species can occur throughout the year, especially when environmental conditions are favorable [98,99]. For example, in urban environments, factors such as poor housing structures provide shelter, while refuse, improper storing of food, and lack of sanitation contribute as major food sources for these rodents [98]. Consequently, these rodents do not show seasonal cycles in such conditions and can breed throughout the year [98,100]. In addition, previous studies showed that Norwegian rats and house mice maintained similar reproductive rates during both dry and rainy seasons [101,102].

Our prediction for the human footprint was met for Norwegian rats, as it contributed more than bioclimatic variables. For example, the distribution expanded and covered other provinces, such as Limpopo, when the human footprint was added to the model. This suggests that the human footprint and bioclimatic variables are significant on the distribution of Norwegian rats [34]. These results support that this species is commensal with humans, and socio-economic factors such as human population density have facilitated its spread [40,41]. For house mice, the human footprint did not contribute much to its distribution; however, this does not suggest that this species is not commensal as it is found in the urban areas [6,103]. Given that Norwegian rat and house mouse are associated with anthropogenic factors, they pose a health risk to humans as they have been reported to carry several pathogens such as toxoplasmosis and leptospirosis [95,96]. These rodent species are also one of the most damaging agricultural pests worldwide, causing millions of dollars in damages and repairs [13,104,105]. In addition, millions of dollars have been spent on managing these rodents in mainland areas and islands [106]. Controlling them in informal settlements can be difficult because of predator ineffectiveness. For example, in South Africa, barn owls (*Tyto alba*) introduced in Johannesburg were not successful in controlling rodent infestation [107]. In addition, domestic cats (*Felis catus*) and dogs (*Canis lupus familiaris*) were found to have little effect in reducing rats and mice populations in urban areas [108].

In most countries and several islands, house mice and Norwegian rats have been associated with the extinction of several native species through competition and predation [98,99]. The introduction of house mice and Norwegian rats is typically associated with the shipping trade in South Africa and other countries [1,6,42]. However, given that these two species are common in the pet trade, it is important that they are not released outside captivity, especially in provinces where they are climatically suitable because they may establish feral populations as they tolerate a wide range of habitats. Svihla [109] encountered a colony of albino pet rats living in a feral condition and interbreeding with wild rats in Honolulu, Hawaii. In South Africa, rats with color patterns typical of laboratory

and pet rats, e.g., black hooded, champagne and albino rats with red eyes, live in feral conditions [110]. Additionally, media reports have implicated pet owners in different countries intentionally releasing or abandoning rats into the wild [111–113]. Consequently, such incidents could be happening in South Africa but are not reported.

The two other mammalian species recorded as invasive in South Africa are European rabbits which are invasive on the offshore islands, and eastern gray squirrels currently distributed in the Western Cape Province [6,78]. The distribution of these two species in South Africa is associated with intentional releases and accidental escapes from captivity [6]. Although European rabbits are invasive on the offshore islands, they have relatively few presence records in the Gauteng and Western Cape Provinces. This indicates that this species is being released or has accidentally escaped captivity. European rabbits and eastern gray squirrels are also regarded among the most destructive mammalian species. For example, European rabbit competes with domestic animals for pasture, and they are also responsible for impacting native species through habitat destruction [114–116]. The eastern gray squirrel negatively affects forestry production, causing millions of dollars in damages and repairs [117,118].

Some of the distribution records for European rabbits and eastern gray squirrels overlapped with their predicted climatic suitability. However, European rabbits and gray squirrels are expected to expand along the coast given their predicted suitability and habitat preferences, including dry areas near sea level and woodland [119–121]. In addition, urban parks, gardens, and agricultural land may inadvertently assist the spread of these species [119–121]. For example, the spread of eastern gray squirrels in South Africa is associated with urban and commensal areas in Western Cape Province [6].

Climate is important in determining the distribution of species [14,15]. However, it is essential to consider that species invasion depends not only on the climate but also on other factors such as high reproductive rate, broad diet, lack of competitors, and predators in introduced areas [44]. For example, the invasive capacity of common marmosets and black-tufted ear marmosets may be explained by their ability to obtain secondary resources in times of scarcity in highly seasonal environments [122,123]. In Brazil, these two marmoset species and common squirrel monkeys have become invasive because of pet escapes and mistaken releases of seized pet animals [42,54]. While these species occur in the same areas in Brazil [42], our results showed that their respective climatic suitability in South Africa was in the coastal areas. Warm temperature, isothermality, dry and wet seasons explain *Callithrix* species' climatic suitability in South Africa. In contrast, common squirrel monkey suitability was explained by temperature seasonality. This indicates that the two marmosets could have higher invasive potential in South Africa compared with common squirrel monkeys. In addition, a study in Brazil also found that the climate for common marmosets and black-tufted ear marmosets was determined by warmer temperatures [123]. Previous studies found that marmosets can occupy and survive degraded habitats because they have a broad diet, social flexibility and are successful breeders [124,125]. Given the advantage of these biological characteristics and a suitable climate, these species pose an invasion risk in South Africa.

Commonly traded species such as Guinea pigs, golden hamsters and winter white dwarf hamsters may pose an invasion risk in South Africa. They are traded in most provinces, and most of their selling localities were within their projected distributions. There is little information on the impacts associated with these three rodent species; the only available literature is on the role of Guinea pigs in the extinction of Laysan rail (*Zapornia palmeri*) in Hawaii through habitat destruction [116]. Another commonly traded species, African pygmy hedgehog, was not predicted to have a large distribution in South Africa. This could be explained by the limited number of distribution records for this species. Therefore, other factors may influence its distribution should it escape or be released from captivity.

Although some of the species were not commonly traded in the pet trade, they were found to have environmental and socio-economic impacts, for example, Mongolian gerbils,

domesticated ferrets and sugar gliders (see [13]). The predicted climatic suitability for Mongolian gerbils was very low, covering only a small portion of the Northern Cape Province. In addition, its present selling points were not within the predicted distribution. As a result, this species does not pose an immediate invasion risk as compared with the others. Given that the occurrence records for sugar gliders are typically in tropical and subtropical environments in its native and invaded ranges [1], the climate may be an important factor in the establishment of this species in South Africa, given that it has large climatic suitability along the coast. In addition, sugar gliders are generalists, and in their invaded range, Tasmania, they negatively affected cavity-nesting bird species populations through competition for nests and predation [126,127]. The bioclimatic variables may also play a role in the distribution of domesticated ferrets in South Africa, with a highly suitable distribution in coastal areas of South Africa, as in their invaded range in New Zealand [128,129]. However, this species could occupy lowlands habitats as it is associated with them in western Europe and New Zealand, where it is invasive [130,131]. Domesticated ferrets may negatively impact the biodiversity in the coastal areas if they become successful invaders in South Africa. In New Zealand, domesticated ferrets became successful invaders because of the lack of predators, and they have been reported to predate on native penguins and ground-nesting birds [132].

## 5. Study Limitations

Studies employing SDMs face several limitations, including their interpretation, efficacy, and data availability [133,134]. Our study is the first to assess the potential range of traded mammalian species based on their occurrences elsewhere in South Africa. Using relatively few records for some species and occurrence records elsewhere may limit the outputs of the models, but using local and more occurrence records may help overcome these limitations and improve the SDM results [135,136]. For example, house mice had larger climatic suitability than the other species in our study as they have extensive local and global records. Behavioral plasticity and local adaptation may result in different actual ranges where these species are released. Therefore, incorporating physiological variables in SDM may be essential in improving the reliability of species distribution [137]. In addition, the ability of species to shift ranges during major climatic changes may affect species distribution and risk of extinction [136,138–140]. For example, Bellard et al. [140] indicated that future climatic scenarios for the "100 of the world's worst invasive species" predicated a consistent shrinking range for amphibians and birds, while terrestrial invertebrates were predicted to increase in their range. The SDMs in the present study were limited as they could not predict if the distribution for these non-native mammalian species would decrease or expand under future climatic scenarios. Therefore, species distribution modeling that integrates future climatic predictions and the phenotypic flexibility of introduced species to persist in these scenarios may be necessary for a greater understanding of their invasive potential [141–143].

## 6. Conclusions and Recommendations

We concluded that all the 14 assessed species have potentially suitable areas in South Africa and may pose an invasion risk should they escape or be released from captivity. Although all the assessed species may need monitoring, the species of most concern are sugar gliders, golden hamsters, winter white dwarf hamsters, house mice, European rabbits, eastern gray squirrels, and Norwegian rats. These species are likely to become invasive or expand their present distribution in South Africa because they have a suitable environment, high availability in the pet trade, and can tolerate wide climatic ranges. In addition, some of the selling points and occurrence records for established species were within highly predicted suitable distribution areas. Additionally, the human footprint contributed the most to the predicted distribution of Norwegian rats. Therefore, this species may further expand its present distribution in South Africa with humans' assistance through the pet trade.

It is important that the current legislation on the sale of highly invasive species with known impacts be revised to protect the South African biodiversity and economy. For example, house mice, Norwegian rats, and European rabbits are listed as category 1b (i.e., invasive species which are strictly prohibited from any form of trade and must be controlled, removed and destroyed) for offshore islands according to the National Environmental Management: Biodiversity Act of 2004 [144]. We recommend that these three species be listed in category 1b for the mainland as they are sold as pets and have large climatic suitability. In addition, Norwegian rats and house mice should be controlled, given that previous studies implicated them as reservoirs for pathogens that can be transmitted to humans in South Africa [95,96,145]. Another species of concern is the eastern gray squirrel which is presently listed as category 1a (i.e., invasive species which must be eradicated and whose any form of trade is strictly prohibited) for KwaZulu-Natal Province and category 3 (i.e., invasive species which may remain in permitted areas or provinces and are prohibited from trade and breeding) for other provinces [144]. Given that this species is already invasive in the Western Cape Province and poses a threat to infrastructure and agricultural production [6,13], it should be listed under category 1a so that it can be eradicated. Species with potential high invasion risk should not be traded in South Africa because of negative impacts reported in other countries [13] which might also occur here.

**Author Contributions:** Conceptualized the study and did the sample design N.S., T.C.S. and C.T.D.; implemented the study N.S.; analyzed the data N.S. and T.C.S.; wrote the draft manuscript N.S.; edited the manuscript before submission T.C.S. and C.T.D.; All authors have read and agreed to the published version of the manuscript.

**Funding:** This research was funded by National Research Foundation (Z.A.; grant number 98404) and the DSI-NRF Centre of Excellence for Invasion Biology, Stellenbosch University (Z.A.).

**Institutional Review Board Statement:** The study was conducted according to the guidelines of the Declaration of University of KwaZulu-Natal, and approved by the University of KwaZulu-Natal—Humanities and Social Research Ethics Committee (protocol code: HSS/0678/018D, date of approval: 5 September 2018).

**Informed Consent Statement:** Informed consent was obtained from all subjects involved in the study.

**Data Availability Statement:** Data for this study is available on request from the authors.

**Acknowledgments:** We are most grateful to the National Research Foundation (Z.A.; grant number 98404) and the DSI-NRF Centre of Excellence for Invasion Biology, Stellenbosch University (Z.A.), for funding. We acknowledge the Ford Wildlife Foundation (Z.A.) for vehicle support and the University of KwaZulu-Natal (Z.A.) for logistic support and funding. We are most grateful for the constructive comments of the reviewers which improved the manuscript.

**Conflicts of Interest:** The authors declare no conflict of interest nor competing interests.

# Appendix A

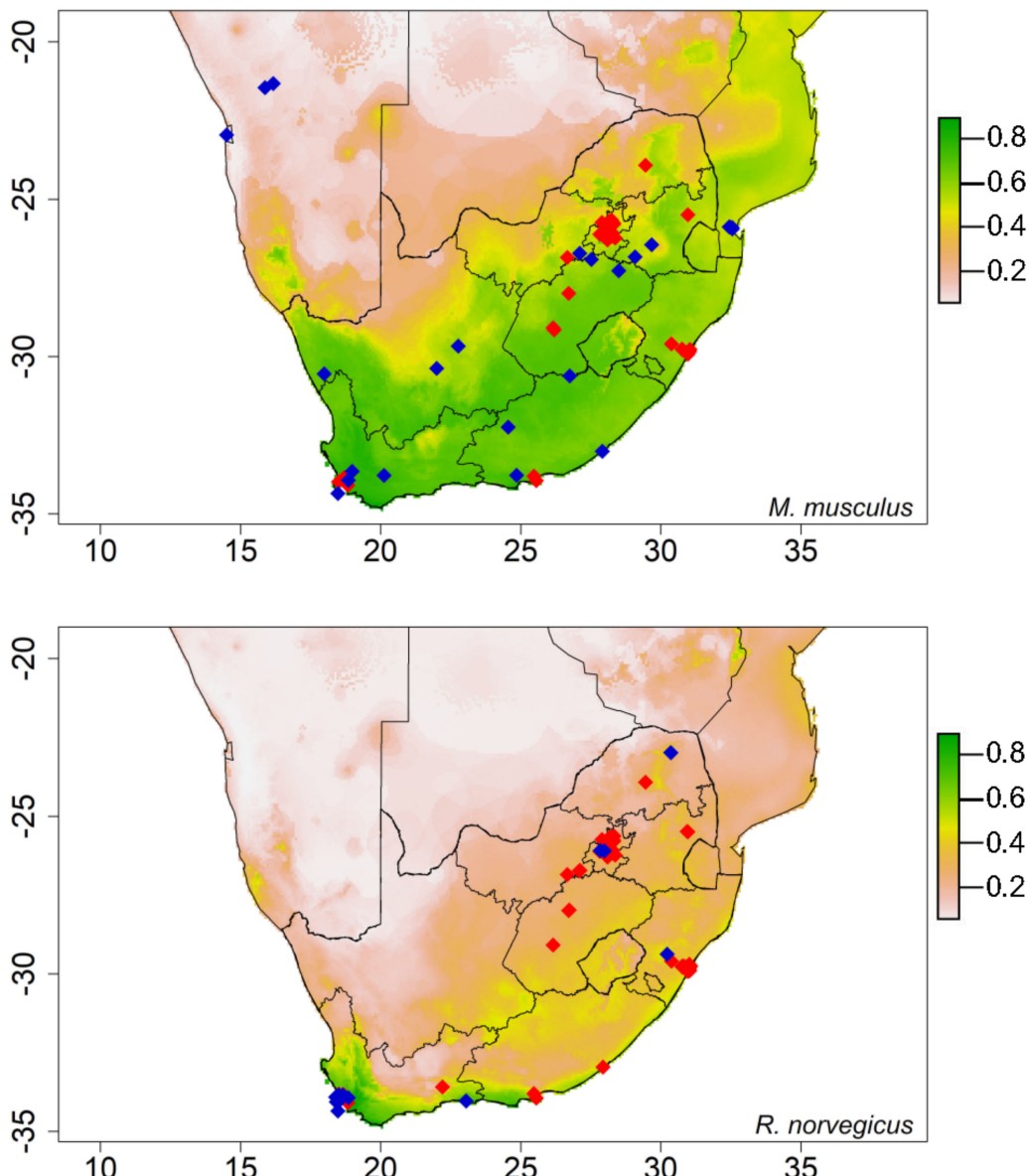

**Figure A1.** The potential distribution from ecological niche modeling of house mice (*Mus musculus*) and Norwegian rats (*Rattus norvegicus*) in South Africa (Note: The color ramp threshold on the right measured the suitability: green indicates the most suitable areas, decreasing to yellow and orange, with light gold and white being unsuitable). Blue dots indicate current distribution records and red dots indicate selling points.

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
