# Peer review of "Predicting the Potential Distribution of Non-Native Mammalian Species Sold in the South African Pet Trade"

_diversity, doi:10.3390/d13100478_

Round 1

Reviewer 1 Report

The manuscript is very interesting and the language is clear.  3 remarks and additions to be made 

  • Line 63-64-65: not only...you also study the effects of human footprint
  • Line 81: "and excluded all pet product stores".....why?
  • In discussion: Line 282-322: a paragraph could be added on the biological characteristics  of the 2 species, Mus and Rattus: reproduction, diet, predator effects, effect of precipitation on life cycle ....

Reviewer 2 Report

Review of Diversity-1351731

Predictie potential distribution of non-native small mammal species sold in the South African pet trade

Dear Authors,

This is an interesting study that takes a novel approach to examining the potential invasiveness of smaller mammals sold in the South African pet trade. It is a companion study to other work on the examining the pet trade in South Africa, and a logical extension of that previously published work. Taken together, manuscripts from this project have great potential to be a catalyst for similar work elsewhere, and this work can be helpful in addressing the issue of alien and invasive species. Thank you for doing this work.

While quite focussed in scope, I found the study generally well done and the manuscript well prepared. That said, I have several substantive issues with the manuscript that I feel need addressed before it is acceptable for publication.

Please find my substantive comments below, followed by those of a more editorial nature.

  1. Overlap with previous papers. While I appreciate that the authors have built upon their companion studies that have been recently published (i.e., Shivambu et al. 2020, 2021), aspects of the this manuscript seem to report some of the same data already published in those previously published works, without acknowledging such. A major recommendation is to make very clear linkages between this manuscript and the other companion studies. It is vital to not report data on the survey of pet shops and those findings as original data from this manuscript; rather focus on the SDM modeling. Ensure to explicitly describe early in the Introduction how this study differs from the others.

  1. Aim(s) not explicit. Related to my comment above, it was not crystal clear to me what the aim of the study was from reading the Abstract or Introduction alone. Please be sure to clearly state what the aim of this work is and why early in the manuscript. The reader should not have to “read between the lines”. This can be accomplished with very minor revision.

  1. Data needs better documented. The data that were used for developing the SDMs is not well described. For example, what was the spatial and temporal extent of the species occurrence records used to develop the SDM? Why were the 19 bioclimatic covariates chosen, specifically? Please add substantially to the Methods what data was used and why, as well as bolstering your description of the model structure used. This is key for the work to be repeatable – and I hope others follow your framework for other locations.

  1. Limitations need addressed. I was surprised that the Discussion did not delve into the limitations of these SDMs. There is a large literature cautioning researchers and managers on the interpretation and utility of SDMs. This does not diminish the value of this manuscript at all; however, limitations do need to be addressed. Hopefully, these can guide future SDM efforts. Top of my mind would be the need to enforce that these results are a first approximation of potential range of these species, based on their occurrences elsewhere. Behavioural plasticity and local adaptation may result in different actual ranges where this species are released. Again, the literature points to several other key limitations of SDMs. Finally, what about climate change? This should be strongly acknowledged here as well.

  1. Wandering Discussion. Unfortunately, I found the Discussion to lack focus and wander. To restructure the Discussion, I suggest you focus on return to your predictions from the Introduction and determine how they are met or not. Do discuss how the SDMs align with what we know about the autoecology of the species studied. As noted above, I would hope to see a meaty paragraph or two about the limitations of the current study (and SDM approach in general).

  1. Weak conclusions. Given the conclusions found in the companion studies, I was a bit surprised that those for this manuscript were comparatively weak. I suggest considering much stronger recommendations to combat the potential for some of these species to become (further) established and invasive in South Africa. I would really like to see some species-specific recommendations for those with comparatively higher potential for becoming invasive in large parts of the country.

Good luck revising your manuscript. I hope to see it soon published.

Sincerely.

Literature Cited:

Shivambu N, Shivambu TC, Downs CT (2021) Non-native small mammal species in the South African pet trade. Management of Biological Invasions 12(2): 294–312.

Shivambu N, Shivambu TC, Downs CT (2020) Assessing the potential impacts of non-native small mammals in the South African pet trade. NeoBiota 60: 1–18.

Detailed and Editorial Comments:

Line 2: Please remove “small” from the title and throughout the text, as most would think of rodents, insectivores, and marsupials less than 1 kg. This study includes many other mammals, including primates and carnivores, so the title is a bit misleading, in my view. However, do clarify that the study only includes mammals >5 kg or so, that are sold as companion animals in the South African pet trade (cite your earlier paper).

Line 12: Replace “them” with “ a subset of 14 species”

Line 15: Delete “Of the recorded species, 14 were selected based on selection criteria”

Line 17: Here and throughout, delete “the” before a species name (e.g., European rabbit). In general, “the” is used far too often in the text, with the result of it being taxing on readers. Please go through the entire manuscript and remove instances were the use of “the” is unnecessary.

Line 21: The concluding sentence of the Abstract is rather weak, in my view. Please consider much stronger recommendations to ensure that small mammals in the domestic pet trade do not become established in the wild. Companion papers by the authors (i.e., Shivambu et al. 2020, 2021) have stronger recommendations than here.

Line 30: Replace “than other vertebrate taxa [6,8]. Mammalian species’” with “and their”. Combine these sentences.

Line 39: Replace “that” with “a”. Delete “The”

Line 43: Replace “The” with “A”

Line 59: Add “in many countries” after “remains unregulated”

Lines 65-67: The first aim was previously done by Shivambu et al. 2020, 2021, so it should not be an aim of this study. Please revise to ensure the companion study results are not repeated here.

Line 73: Please see the comment above. Much of this sub-section does not seem to be presenting a new analyses. I believe that this work (i.e. “data collection”) was largely reported already in Shivambu et al. 2020, 2021. As such, please state such and cite those previous studies here.

Line 77: Delete the sentence beginning with “We developed the map…” Not necessary.

Line 95: This is not a new sub-section. It should be combined with that above.

Lines 99-105: These lines should be part of the next sub-section on SDM.

Line 100: Replace “modelling” with “models”. Add “for each species” after “occurrence records”

Line 103: Its not clear to me from where the occurrence records were drawn from. That is, was a global dataset used, or more regional. This is key information that is missing.

Line 108: Replace “of the 14” with “selected”

Line 109: Why were these 19 bioclimatic variables chosen, specifically?

Line 135: Insert “potential” before “range”

Line 149: Delete “ and therefore, their results were presented”

Table 1 needs some more thought. Adding Order and organizing the table by Order would be useful. I don’t think the columns to the right of “availability” are necessary and could be deleted. Table A1 should be combined with this table – BUT be sure to cite the previous companion papers that first present these data.

Line 213: Typo – “Fiure”

Line 221: Rephrase to “For the sugar glider, bio 3 and 14 contributed most to the model (Table A2).”

Line 230: Replace “lowest” with “smallest”

Line 319: Delete “, biting human babies and elders in Alexandra Township”. Unnecessary details.

Line 365: Add “competitors and” before “predators”

Figure A1 and A2 contribute little and should be deleted. Reporting these results as text is sufficient.

Reviewer 3 Report

Dear authors,
I was looking forward to reading the manuscript presenting the results of your study on the potential distribution of non-native small mammal species involved in South African pet trade. The study shows that the non-native small mammal species which are commonly traded in South Africa, have large potential distribution areas in this country, what makes them important potential invaders. Although the study does not contain any novel ideas or earth-shattering results, this is a topic and study of potential interest in the current context, with practical implications, therefore it is worth considering for publication. However, there is room for significant improvement, both in content and form.
The main aim of the paper is to predict potential future distribution of non-native species based on climatic conditions, but climate is changing rapidly at the moment. Thus, climate scenarios should be included in the analyses and the effect of climate change need also to be discussed.
Introduction
The aims of the study should be better presented. In the present form, aim 2) is a bit misleading, because the invasion history and pathways have nothing to do with your study, they were not established based on your data (or at least it appears so), but they were taken from the literature, so you can not include this as an aim. The aim(s) of the study should be made clear also in the abstract.
Material and Methods
The map shows very clustered sampling points (both pet stores and online sites). How where they selected? Is there a logic behind this clumped pattern? If not, may this be a source of bias in your results? I think you should address this issue.
Results
Do you also have the number of individuals available in different pet stores and online sites? If yes, you should report it (probably they are correlated, but not necessarily), as not only occurrence but also abundance of individuals increases the risk of release or escape from captivity and thus, the invasion potential.
Figures 2, 3, 4 and 5 have essentially the same content – illustrating potential distributions. I would group them in one panel. And the maps for the mouse and the rat are redundant. You should keep those with the human footprint and remove or move the ones with only the climatic variables in the annexes, as they have only theoretical value, as the anthropic factor has a significant effect and its existance can not be ignored.
When you refer to the maps indicating the provinces, these need to be identified, something that is not possible looking only at the distribution maps. You need to add numbers for each region and explain then in the figure caption.
2
The results section is too wordy and difficult to follow. Instead of so many words, it would be more efficient to bring tables A2 and A3 (which are main results) to the main text and let the reader to read the values, if interested. This would increase the flow and digestibility of the results section.
Discussion
This section feels a little diluted and difficult to follow. You should try to focus on the main messages. In addition, referring to the region names does not tell much to the non-native reader, so you should say something about their climate, human population density and pressure, status of invasive species of other taxa etc.
Minor comments
Lines 30-31 – you either use comparative – ”Mammalian species are more successful invaders worldwide than other vertebrate taxa”, or superlative – ”Mammalian species are among the most successful invaders vertebrate taxa worldwide”.
Line 40 – change to ”species distribution modelling”.
Line 69 – ” extensive occurrence records” you mean worldwide?
Lines 96-97 – ”…we excluded ten species based on the following criteria; 1) popular in the South African pet trade…” as it is, one may understand that the popular species were not included in the analysis.
Lines 99-100 and elsewhere – use SDM instead of ”species distribution modelling”, as you already defined the abbreviation, and should use it consistently afterwards.
Line 114 – why ”nine or ten variables”? In table A2 there are 11.
Table 1 – in the last column I would rather include only the reference number and give the site link in the reference list.
Oryctolagus cunniculus – ”All continents” or ”All continents except Antarctica”?
Line 177 – Bio 2 and Bio7 are not explained.
Line 218 – Bio 4 was already explained.
Line 230 – change ”lowest” with smallest or most reduced
Line 284 – ”the latter two” - not the rabbit too?
Lines 286-288 – This is results.
Lines 290-292 – idem
Line 306 – ”These rodent species” is redundant. Change to they.
Lines 340-341 - ” relying on the climate to survive” – this sounds strange. You need to explain what you mean.
Line 377 – ”these species” – it is not very clear to which species you refer to. Name them.
Line 383 – add ”risk” to ”invasion”
Line 383 – delete the second comma
Line 403 – add ”be” to ”released”
Line 416 – add ”are” to ”no longer”.

Reviewer 4 Report

Manuscript ID: diversity-1351731
Predicting the potential distribution of non-native small mammal species sold in the South African pet trade

Review

In general, manuscript is of considerable interest for presenting future aspects of the mammal diversity in South Africa. Pet trade and subsequent release or escape is the most possible route of establishing populations of many exotic species, therefore analysis of the suitability of environment for these species is appropriate.

I just think, that for the most scientists definition of the “small mammal” does not include rabbits and other similarly sized species. Some of the medium-sized species (e.g., fox, ferret, capuchin) are included into the Table A1. Please add explanation, and the references, to the Introduction or the Table.

Abstract

Line 12: please consider “total of 24 small mammal species sold and selected 14 of these for the further analysis based on their popularity…”

Line15: delete “Of the recorded species, 14 were selected based on the selection criterion.”

Lines 18–19: Synantropic are two species, house mouse and Norway rat, not only the latter one. Authors may find references for the other countries. This is only small mistype, as human footprint modelling was done for both species (Figure 3), however it needs correction.

Introduction

Please refer specifically to introduced mammal in the South Africa, from [5]

Line 30: not “than”, maybe “among”?

Lines 59–60: unregulated not in all countries. If authors refer to Africa, please state this clearly. If not – give reference for the other parts of the world. Many countries have at least some regulations for the trade of (potentially) invasive species.

Line 66: add “with the aim to determine”

Material and methods

Line 77: delete “(Figure 1)”

Line 81: delete “and excluded all pet product stores” as mistype

Lines 81–83: please consider edit “We compiled a list of traded non-native small mammalian species and recorded number of pet stores where these species were sold (Table A1).”

Lines 85–91: needs editing of the style, i.e. “few Provinces (<4)” could be just “in less than four provinces”, and so on.

Figure 1: quality is too low, dots have shadow

Lines 96-99: not clear. For example, if species is “popular in the South African pet trade” – it was excluded or not excluded from the list, and so on. Please make this clear.

Results

Line 138: small and medium-sized mammals?

Line 146: there is a difference between (1) “invaded countries” and (2) “countries, where species is listed as invasive” – number of the latter is much bigger that you show in the Table 1. Make corrections, and show references for the latter statement, if you will use it.

Figure 2: please make the bar wider, colours are hardly seen. Also please consider colour selection of the dots; blue and black are hardly conspicuous now.

Lines 174–176: no need to give all AUC values, as these are in the Table A2.

The same throughout the rest of Results text – do not list values presented in the Tables.

Line 184: Tables A3, A4; the same throughout the text

Line 190: Figures 2, 3; the same throughout the text

Figure 4: please make the bar wider, colours are hardly seen.

Figure 5: please make the bar wider, colours are hardly seen.

Please consider shortening of the Results text – Table A2 should be moved to the main text, and bioclimatic variables then may be omitted from the species-related text, referring to the Table. Please also explain why some values in the Table A2 are given in the fold font.

Discussion

Line 339–341: delete the last sentence

Otherwise text is good, I appreciate it and the references.

Conclusions

Most of the text presented here is not conclusions – it is recommendations. Please consider changing chapter title accordingly.

Appendixes

See above

References

I appreciate choice of references, however, few more will be added (see above)

Round 2

Reviewer 3 Report

Dear authors,

I was happy to receive for review an improved version of your manuscript, with many of the reviewer’s comments and suggestions adressed. However, I feel that the last part of discussion, regarding limitations of SDMs and expansion of the study including climate change scenarios should be further developed, with references to other studies that address these issues.

Some other minor comments and suggestions are given in the attached document.
